# The Effect of Mass Testing, Treatment and Tracking on the Prevalence of Febrile Illness in Children under 15 in Ghana

**DOI:** 10.3390/pathogens11101118

**Published:** 2022-09-29

**Authors:** Collins Stephen Ahorlu, Ignatius Cheng Ndong, Daniel Okyere, Benedicta A. Mensah, Chuo Ennestine Chu, Juliana Y. Enos, Benjamin Abuaku

**Affiliations:** 1Department of Epidemiology, Noguchi Memorial Institute for Medical Research, College of Health Sciences, University of Ghana, Legon, Accra P.O. Box LG 581, Ghana; 2Department of Biochemistry, Faculty of Science, Catholic University of Cameroon, Bamenda P.O. Box 572, Cameroon

**Keywords:** malaria, febrile illness, Pakro, Ghana, malaria prevalence, MTTT

## Abstract

Background: Malaria remains a serious threat to children under 15 years of age in sub-Sahara Africa. Mass testing, treatment and tracking (MTTT) of malaria has been reported to reduce parasite load significantly. However, the impact of MTTT on the prevalence of febrile illnesses in children under 15 is not yet clear. This study explores the impact of MTTT complemented by prompt home-based management of malaria on febrile illnesses and their treatment in children under 15 years old. Methods: A cohort of 460 children under 15 years were recruited from the Pakro subdistrict in Ghana during a community-wide implementation of a quarterly MTTT intervention. The MTTT implementation involved testing all household members for malaria using RDTs, and positive cases were treated with Artemisinin-based combination therapy (ACT). Febrile illnesses among this cohort in the two weeks prior to the prevalence survey at baseline and endline were recorded to constitute date for analysis. Results: The prevalence of febrile illnesses, such chills, convulsion, fever, diarrhoea, headache, vomit, cough/rashes or stomachache, etc., were recorded). Asymptomatic parasitaemia prevalence at baseline was 53.3%, which dropped to 44.1% at evaluation. An overall decrease in the parasitaemia prevalence of 33.0% (OR = 0.67, CI = 0.50, 0.89) was observed at evaluation compared to baseline after adjusting for age, ITN use and temperature. A 67% decrease in severe anaemia cases (Hb < 7) was observed at evaluation. Conclusion: Our findings suggest that implementing MTTT complemented by home-based timely management of malaria does not only reduce febrile illnesses and for that matter malaria prevalence, but could also reduce severe anaemia in children under 15 years old.

## 1. Introduction

Febrile illnesses in children are the most common cases presenting in health facilities [1,2], and over the last two decades, their prevalence has decreased significantly. The reduction can be attributed to changes in infectious diseases patterns, mainly because of efforts in malaria control strategies as well as the introduction of malaria vaccines [3]. However, febrile illnesses remain a major public health challenge, especially for children in resource-limited countries in sub-Saharan Africa where the prevalence of infectious pathogens is high. It is estimated that the number of febrile episodes in children under 5 ranges from 2 to 7 episodes per person per year, accounting for 80% of patients presenting acute medical conditions in health facilities in malaria-endemic countries [4,5,6]. 

The symptomatic overlap between febrile diseases impedes their diagnosis on clinical grounds and poses a considerable challenge for health professionals and surveillance systems. In malaria-endemic settings, malaria is usually the predominant cause of fever among children. However, non-malaria causes of fever include other pathogens—including parasites, viruses and bacteria—each of which requires a different type of treatment and management [5]. A combination of factors may influence the incidence and aetiology of infectious febrile illnesses, including the tropical climate, multiple diseases at presentation, innate or acquired host factors (micronutrient deficiency, immunodeficiency, comorbidities), insufficient lack of resources (low income) to procure medicines and vaccines or other preventive tools, lack/misuse (due to lack of training or information) of suitable diagnostic tools, increase in pathogen resistance to antimicrobial medicines, and social and community factors [5]. 

Currently, there are ongoing efforts to scale up malaria control interventions to further reduce malaria burden in areas where it is still a public health problem [7,8]. There has been a concerted effort in the past decade to evaluate the possible effects of massive parasite clearance using intermittent preventive treatment (IPT) in children under five years old in different parts of Africa [9,10,11]. Artesunate- and Amodiaquine-based IPT in children under 15 combined with home-based management of malaria by community workers has been demonstrated to clear more than 90% of the parasite load in children under five years [1]. Mass test, treat and track (MTTT) has also demonstrated a 24% reduction in parasite carriage in all age groups [12]. In addition to reducing parasite carriage and malaria-related fevers, it is important to assess whether MTTT can also reduce other febrile illnesses and all-cause anaemia in children under 15 years of age. This paper attempts to assess the effect of MTTT on the prevalence of febrile illnesses and all-cause anaemia in children under 15 years of age in the Pakro subdistrict of Ghana. 

## 2. Methods

### 2.1. Study Participants 

This study is part of a larger study of which, the implementation involved delivering of MTTT three times in a year combined with home management of malaria in the entire population in the Pakro subdistrict of Ghana. Community entry activities to sensitize the chiefs and the general population were conducted at the beginning of the study through meetings and community durbars (a gathering of community members for discussions of matters affecting them that is often called for by community leaders) [1]. Following the acceptance of the project by community leaders and the population, every household was given a number and all members were registered. All households were given a unique identification code. Parental consent was obtained from caretakers of the children. In addition, individual assent was obtained from adolescents aged 12 to 14 years old before they were enrolled in the study.

### 2.2. Study Design

The design and implementation scheme for this study has been described in detail in a previous publication [13]. Briefly, two weeks prior to each MTTT intervention, a survey was conducted to determine the anaemic and febrile status of children recruited in this study group. July 2017 and July 2018 surveys constituted the baseline and the evaluation, respectively. All participants were tested by CBHVs, and all malaria-positive cases were treated with Artesunate–Amodiaquine (ASAQ) or Artemether–Lumefantrine (AL), as reported in [13]. The use of these regimens depended on what the national malaria control programme was able to supply at each time point, since both combinations have been approved for the treatment of uncomplicated malaria in Ghana. The CBHVs administered the first dose, and parents administered the second and third doses. The CBHVs performed follow ups to ensure compliance. The drugs were administered after meals.

### 2.3. Questionnaire Administration 

Seven days prior to each intervention, all caregivers and parents of the selected children in the subgroup were surveyed using a questionnaire designed to solicit information on febrile illnesses and the use of an insecticide-treated net. Other information captured in the questionnaire included age, sex and febrile symptoms exhibited in the two weeks prior.

### 2.4. Timely Treatment of Suspected Febrile Malaria Cases in the Community

To facilitate home-based management (HbM) of malaria, two community-based health volunteers (CBHVs) from each community were recruited, trained and provided with the protocol, rapid diagnostic tests (RDTs) and artemisinin-based combination therapies (ACTs). The CBHVs provide community-based timely testing of febrile (suspected malaria) cases. Those confirmed to be carrying the malaria parasite (Malaria RDT-positive) were promptly treated using ACTs, and those found to be negative were referred to the health facility for further evaluation and treatment. This activity was conducted through biweekly home visits to record all febrile cases two weeks before implementing MTTT.

### 2.5. Sample Size

To determine any effect of the intervention on the children aged 6 months to 14 years, a community survey with a sample of 368 children was needed to determine the malaria prevalence, which was projected at 50% in the study population. The sample size was determined using the formula of Yamane, where *n* = *N*/[1 + *N*(*e*^2^)] [14], considering a 95% confident level (CI) and ±5% precision. Assuming a loss to follow up of 10% and a non-response rate of 10%, the sample size was readjusted to 460 children. Children recruited to the study were classified into two age groups: 1–4 years and 5–14 years.

### 2.6. Inclusion Criteria

To be included in the longitudinal cohort study, participants had to be aged 6 months to 14 years and be a resident in the study area, at least at the time of recruitment, and have no immediate plan to relocate outside of the study area. Willingness to participate was evident upon completion and signing of a consent form by the caretaker of each child and an assent form by children aged 12–14 years. 

### 2.7. Exclusion Criteria

Any child whose caretaker planned to relocate outside the study area or had a life-threatening illness (excluding malaria) was not recruited for this study.

### 2.8. Data Collection 

In order to facilitate tracking, a community register was developed as previously described [13]. Briefly, all households were numbered, and everyone had a unique number which was linked to a particular household. Communities were divided into neighbourhoods, and CBHVs were assigned a specific area where they are well-known. At times, the CBHVs took appointments with household heads before the visit, especially with those who spend a lot of time outside the home. In some instances, the CBHVs had to visit the homes several times to be able to attend to the participants. 

Following consent, blood was drawn from a finger prick. At baseline, all available participants were tested for the presence of malaria parasites using RDTs (prevalence survey) before treatment (intervention). To test for anaemia, a portable automated Hemocue photometer (URIT Medical Electronic Co., Ltd., Guangxi, China) was used to determine the concentration of haemoglobin (Hb). Anaemia in this study was defined as Hb levels less than 10 g/dL [1]. Children with severe anaemia (Hb less than 7 g/dL) were referred to the Pakro Health Centre for follow up. 

### 2.9. Data Management and Analysis 

Data were analysed using STATA (version 15). The unit of enrolment was the household. Malaria prevalence was reported as proportions of participants confirmed during screening to be carrying the malaria parasite, and they were stratified by subject characteristics, including age and sex. A chi-square statistic was used to compare prevalence of parasitaemia across age groups, gender and reported febrile illnesses at the 95% confidence level (*p* = 0.05). Univariate and multivariate regression analyses were employed to determine the strength of the relation between variables.

## 3. Results

### 3.1. Mass Screening Coverage

A total of 460 children, 237 (51.5%) females and 223 (48.5%) males, were selected for the cohort study. Four MTTT interventions were conducted from July 2017 to July 2018. Coverage at baseline (July 2018) was 432/460 (93.9%) and 356/460 (77.4%) at evaluation (July 2018). At the time of the surveys, especially in 2018, 103 of the children were not available for testing because they had temporarily travelled or relocated outside of the study area with their caretakers, and only 1 child (0.2%) had refused to be tested. 

### 3.2. Prevalence of Asymptomatic and Symptomatic Parasitaemia 

The prevalence of asymptomatic parasitaemia (AP) at baseline in July 2017 was 230/432 (53.26%), but at evaluation in July 2018 it dropped to 157/356 (44.1%) (*p* = 0.011) (Table 1). The reduction in AP after a year-long intervention was significant in females (52% to 39%, *p* = 0.007) compared to males, where there was no significant reduction (54% to 50%, *p* = 0.395) (Table 1). A significant decrease in AP was observed in the age group of 5–14 years (*p* = 0.012), while the decrease was not significant for the age group of 1–4. In addition, the prevalence of AP in children who reported to have slept under an insecticide-treated net (ITN) reduced significantly by 10% from baseline to July 2018 (*p* value = 0.014). 

The MTTT intervention reduced AP by 33% from 230/432 (53.26%) in 2017 to 157/356 (44.1%) in 2018 after adjusting for age, temperature and the use of an ITN (odds ratio (OR) = 0.67, CI = 0.50, 0.89, *p* value = 0.006) in multivariate logistic regression (Table 2). The mean temperature in children with AP increased by 26% compared to the temperature in children without AP (OR = 1.26, CI = 1.05, 1.51, *p* value = 0.006). However, age and ITN use were not statistically significant predictors of malaria parasitaemia after adjusting for confounders (OR = 1.02, CI = 0.98, 1.06, *p* value = 0.244 for age and OR = 1.30, CI = 0.89, 1.90, *p* value = 0.114 for ITN use). 

### 3.3. Symptomatic Malaria Prevalence

Symptomatic malaria prevalence in children under 15 years old decreased from 55.1% at baseline to 50.7% at evaluation, a 4.4 percentage point reduction, which translates to a 7.98% proportional drop, albeit the reduction was not statistically significant (*p* value = 0.567). 

### 3.4. Febrile Illnesses among Children under 15 Years Old

The study showed a general reduction in the trends of all-cause febrile illnesses, captured as chills, convulsion, fever, diarrhoea, headache and vomiting, among others, over the study period. It is worth stressing that fever reduced by 43.45% (from 38.2% to 21.6%). Headache, which was reported by 39.6% at baseline and 33.7% at evaluation, emerged as the most common febrile illness reported in children under 15 years of age in the study area (Table 3). 

There was a significant difference between children with RDT-confirmed parasitaemia-positive cases reporting diarrhoea at baseline (July 2017) and evaluation (July 2018). Though there was a reduction in the number of RDT parasitaemia cases reporting chills, fever, headache and vomiting, the differences between the two time points were not significant (Table 4). Analysis of the febrile illnesses showed that fever, headache, vomit, anaemia and convulsion were significantly associated with malaria parasitaemia (*p* value < 0.05), while chills, diarrhoea and others, such as cough/rash or stomachache, were not (*p* value > 0.05). Fifty-five percent of children with fever, 51% of those who reported headache, 55% of those who vomited and 64% of those with severe anaemia also tested positive for malaria. 

### 3.5. Prevalence of Anaemia in Children under 15 Years

At baseline, 25 (7.0%) of the children were severely anaemic (Hb < 7.0), and 116 (32.3%) were moderately anaemic (Hb = 7.0–9.9). At evaluation, the proportion of severe anaemia dropped significantly from 8.0% at baseline to 4.4% at evaluation (X^2^ = 16.2, *p* < 0.001). Moderate anaemia dropped from 43.5% at baseline to 23.4% at evaluation (X^2^ = 14.2, *p* = 0.001) among children under 5 years of age. Consequently, the proportion of non-anaemic children in the study increased from 60.7% to 77.8% at evaluation (Table 5). These findings suggest that implementing MTTT in the community could contribute enormously to the reduction in all-cause anaemia in children aged 6 months to 14 years. 

## 4. Discussion 

### 4.1. Coverage

Over the one-year period (July 2017 to July 2018), coverage among the children in the cohort study was more than 86% at baseline and gradually reduced over time till it reached 77% at evaluation. Population migration, involving the relocation of entire families out of the study area, was the main contributing factor in the decrease in coverage. This contrasted with the findings by Cook and colleagues [15], where a decrease in coverage was due mainly to absenteeism or a high refusal rate. Community engagement, as evident by the number of signed informed consent forms received, was very high among the caretakers in our study. This could have been stimulated by the engagement with the local chiefs through durbars at the planning and implementation stages, and continuous sensitization of the population during the study. It could also be partly due to the involvement of community volunteers, who were familiar faces to the caretakers of the children [12]. 

### 4.2. Prevalence of Parasitaemia

This study evaluated the effect of malaria MTTT complemented by HbM on the prevalence of febrile illnesses in children aged from 6 months to less than 15 years in the Pakro subdistrict of Ghana. The results suggest that within a year of the intervention, the prevalence of malaria asymptomatic parasitaemia (AP) decreased by 33% at evaluation when compared to baseline. This paper presents findings from a point prevalence and therefore did not consider seasonal variations, which will need to be compared over a longer time. 

The World Health Organization recommends that malaria diagnosis and treatment be deployed as part of integrated management of fevers, based on WHO algorithms available for different age groups and levels of care [6]. The use of ACTs in MTTT of malaria could be of immense benefit to the health of children under 15 years of age if implemented at a large scale in endemic areas [16]. In our study, there was an overall decrease in the number of parasitaemia-positive cases in the study population when compared to baseline (from 230 cases at baseline to 157 cases in July 2018). The observed reduction in parasitaemia in this study is similar to recent findings in Ghana, which suggests that combining intermittent preventive treatment in children (IPTc) combined with HbM of malaria is effective in reducing parasitaemia by more than 90% in children under five and sustaining low levels over a two-year period [1,17]. On the other hand, the finding is in contrast with earlier reports by Burkina Faso [18] that mass testing and treatment of asymptomatic parasitaemia did not reduce parasitaemia prevalence. Despite the reduction in parasitaemia, the time lapse between interventions (four months) in this study was long enough to allow for re-establishment of the parasite population as the ACTs used afforded limited post-treatment protection for between 14 and 28 days [19,20]. Shortening the time between interventions from four months to two months could lead to the clearing of submicroscopic parasite levels and further reduction in the potential for transmission [13]. The high prevalence of parasitaemia in the older children compared to younger ones corroborates earlier findings in Kenya revealing that the age group with the highest prevalence of malaria may be shifting from children under 5 to between 5 and 10 years [20].

### 4.3. Prevalence of Febrile Illnesses 

Fever case management remains a major barrier to improved child health worldwide, despite the availability of low-cost and reliable child survival technologies [21]. The usefulness and effectiveness of community-based treatment of disease conditions such as diarrhoeal diseases and malaria have been demonstrated [22]. It must be noted that the frequency of febrile illnesses may, however, be influenced by seasonality, as reported in earlier studies [23,24,25]. 

There have been extensive studies on febrile illnesses and overuse of antimalarials in the treatment of febrile illnesses [21,22,26,27,28]. However, few studies have focused on how the use of antimalarials protects against non-malarial febrile illnesses. Our study demonstrated that treatment of asymptomatic children with ACTs led to a reduction in all-cause febrile illnesses that share symptoms, such as fever, chills, headache, diarrhoea, convulsion, and vomiting. The contributions of antimalarials in reducing febrile illnesses may be further studied and exploited as a protective intervention against febrile illnesses in general, though other factors such as antimalarial drug resistance should be considered.

### 4.4. Effect of ITN Used

The fluctuations in the proportion of caregivers who reported the use of an ITN the night before the survey suggest that ITN use may be seasonally driven [29]. This suggests that people remember the necessity of using an ITN when they perceived a surge in the mosquito population or when more children come down with febrile illnesses. A high proportion of caregivers reported using the ITN in July 2017, which is peak transmission season and coincidentally the peak period of febrile illnesses reports. Since mosquito nets are not used consistently, any renewed quest to use them may not really consider whether the nets are in good condition, and this may explain why individuals may use ITNs, which are not effective when they are torn with one or several holes [30] or the insecticide is covered by dirt to the extent that it cannot be effective.

### 4.5. Prevalence of Anaemia

In this study, we considered all-cause anaemia and attempted to separate severe anaemia (Hb < 7 g/dL) from moderate anaemia (Hb = 7.0–9.9 g/dL). The observed declines of 67% in severe anaemia and 8% in moderate anaemia suggest that implementing MTTT complemented by HbM of malaria could improve the anaemic situation in children under 15. Our findings are in line with earlier reports in Ghana which suggest that combining IPTc with home-based management of malaria could reduce all-cause anaemia in children [1,17,25]. These findings further strengthen the observation that the use of ACTs in Ghana could improve the situation of anaemia in children under 15 [25]. Apart from referring the children to the health facility, the research team encouraged caretakers of such children to improve their dietary intake with cocoyam and cassava leaves, which are rich in iron, through community sensitization sessions. At the health facility, there exists an ongoing nutrition supplement programme for severely anaemic and undernourished children. However, most of the children identified as anaemic could not benefit from the programme. This is because a child can only benefit if the arm measurement using the tape falls within the red segment, as recommended by the WHO. We therefore recommended the nutrition supplement programme include anaemic children using the Hb measurement. 

### 4.6. Limitations of the Study

There were some drawbacks to the analysis. While most of the data were taken from structured exit interviews with caregivers of febrile children, this may not reflect the true impact of the intervention on other febrile illnesses considered [27]. However, some of the data focused on structured findings, including symptoms reported and malaria testing using an RDT. Some malaria cases might have been missed by the RDTs. Additionally, the contributions of the MTTT and HbM were not independently assessed during this study. It is likely that certain answers of the caregivers could have been influenced by the way the community volunteers interviewed them. It was, however, assumed that the responses obtained were correct. 

## 5. Conclusions

Our findings suggest that implementing MTTT complemented by home-based timely management of malaria through community volunteers does not only reduce febrile illnesses and for that matter malaria prevalence but could also reduce severe anaemia in children under 15 years old. Further studies are required to fully document the impact of this reduction across seasons to draw meaningful conclusions. We recommend that the first year of MTTT interventions for malaria be implemented at intervals of two months instead of the four months in this study to bring down the parasite reservoir drastically. This will help reduce the effect of seasonality on the prevalence of parasitaemia in children under 15 and further reduce the burden of malaria on the family, especially those in sub-Saharan Africa. The study further demonstrates that community health workers, when trained, could be very instrumental in home-based management of malaria. 

## Figures and Tables

**Table 1 pathogens-11-01118-t001:** Univariate analysis of impact of intervention on prevalence of asymptomatic malaria.

Characteristics	Baseline Survey, n/N July 2017 (%)	Endline Survey, n/N July 2018 (%)	χ^2^ Value	*p* Value
Asymptomatic Prevalence	230/432 (53.26)	157/356 (44.1)	6.5	0.011
Sex				
Male	112/207 (54.1)	84/169 (49.7)	0.7	0.395
Female	118/225 (52.4)	73/187(39.0)	7.4	0.007
Age group (years)				
<1	2/9 (22.2)	2/5 (40.0)	0.5	0.48
1 to 4	71/152 (46.7)	51/132 (40.2)	1.2	0.266
5 to 14	157/271 (57.9)	102/219 (46.6)	6.3	0.012
Uses ITN				
No	34/71 (47.9)	28/67 (41.8)	0.5	0.472
Yes	196/361 (55.4)	129/289 (44.6)	6	0.014

**Table 2 pathogens-11-01118-t002:** Impact of MTTT on asymptomatic malaria in children under 15 years old.

Characteristics	Unadjusted OR (CI)	*p* Value	Adjusted OR (CI)	*p* Value
Timeline				
July 2017 survey	Ref	0.009	Ref.	0.006
July 2018 survey	0.69 (0.52, 0.92)		0.67 (0.50, 0.89)	
Age group (years)				
<1	Ref		Ref	<0.001
1 to 4	1.36 (0.60, 3.09)	0.456	2.10 (0.64, 6.91)	0.22
5 to 14	1.61 (0.71, 3.64)	0.245	3.08 (0.95, 10.03)	0.06
Mean ambient temperature	1.19 (1.05, 1.35)	0.007	1.26 (1.05, 1.51)	0.012
Uses ITN				
No	Ref	0.172	Ref.	0.114
Yes	1.17 (0.94, 1.45)		1.35 (0.92, 1.98)	

**Table 3 pathogens-11-01118-t003:** Prevalence of febrile illnesses among children under 15 years.

Variables	Baseline (n = 455) N (%)	Evaluation (n = 356) N (%)
Chills	108 (23.74)	2 (0.6)
Convulsion	4 (0.9)	-
Fever	174 (38.2)	77 (21.6)
Diarrhoea	75 (16.5)	13 (3.7)
Headache	180 (39.6)	120 (33.7)
Vomiting	55 (12.1)	28 (7.9)
Others (cough, rash or stomach-ache)	164 (36.0)	61 (17.1)

**Table 4 pathogens-11-01118-t004:** Association of febrile illness with confirmed RDT-confirmed parasitaemia at different time points.

Characteristics	RDT-Positive at Baseline July 2017	RDT-Positive at Evaluation July 2018	χ^2^ Value	*p* Value
Chills				
No	177/327 (54.1)	155/354 (43.8)	7.3	0.007
Yes	52/104 (50.0)	2/2 (100)	2	0.161
Convulsion				
No	227/429 (52.9)	157/356 (44.1)	6	0.014
Yes	3/3 (100)	0/0 (100)		
Diarrhoea				
No	195/361 (54.0)	154/349 (44.9)	3	0.129
Yes	35/71 (49.3)	3/13 (23.1)	5.9	0.016
Fever				
No	140/266 (52.6)	115/279 (41.2)	7.1	0.008
Yes	90/166 (54.2)	42/77 (54.6)	0	0.962
Headache				
No	137/262 (52.3)	95/236 (40.3)	7.2	0.007
Yes	93/170 (54.7)	62/120 (51.7)	0.3	0.609
Vomiting				
No	199/378 (52.7)	142/328 (43.3)	6.2	0.013
Yes	31/54 (57.4)	15/28 (53.6)	0.1	0.74
Others (Cough/rash or stomachache, etc.)				
No	143/278 (51.4)	138/295 (46.8)	1.2	0.265
Yes	87/154 (56.5)	19/61 (31.1)	11.2	0.001
Anaemia				
Non-Anaemic (Hb ≥ 10)	118/218 (54.1)	107/277 (38.6)	11.8	0.001
Moderately Anaemic (Hb = 7.0–9.9)	61/116 (52.6)	44/71 (62.0)	1.6	0.209
Severely Anaemia (Hb < 7.0)	16/25 (64.0)	6/8 (75.0)	0.3	0.687

**Table 5 pathogens-11-01118-t005:** Prevalence of anaemia in children across different age groups.

Age	Anaemia Status	Baseline	Evaluation	Pearson χ^2^ Value	*p* Value
<5 years	Severely Anaemia (Hb < 7.0)	11(8.0)	6 (4.4)	16.2	<0.001
	Anaemic (Hb = 7.0–9.9)	60 (43.5)	32 (23.4)		
	Non-Anaemic (Hb 7–9.9)	67 (48.5)	99 (72.2)		
5–14 years	Severely Anaemia (Hb < 7.0)	14 (6.3)	2 (0.9)	14.2	0.001
	Anaemic (Hb = 7.0–9.9)	56 (25.3)	39 (17.8)		
	Non-Anaemic (Hb 7–9.9)	151 (68.3)	178 (81.3)		
All Age groups	Severely Anaemia (Hb < 7.0)	25 (7.0)	8 (2.3)	26.6	<0.001
	Anaemic (Hb = 7.0–9.9)	116 (32.3)	71 (19.9)		
	Non-Anaemic (Hb > 9.9)	218 (60.7)	277 (77.8)		

## Data Availability

The data analyzed are presented in the paper; however, the dataset is available in the Department of Epidemiology, Noguchi Memorial Institute for Medical Research, University of Ghana, and can be made available by the corresponding author upon reasonable request.

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
