# Peer review of "The Effect of Mass Testing, Treatment and Tracking on the Prevalence of Febrile Illness in Children under 15 in Ghana"

_pathogens, 2022, doi:10.3390/pathogens11101118_

Round 1

Reviewer 1 Report

The authors present the results of MTTT interventions conducted in 2017-2018 in 460 Ghanaian children in Pakro, Ghana. The present study is actually part of a larger study, the results of which were published in 2019 (REF 12 and 21). The effects of ACTs used in MTTT on asymptomatic carriage and symptomatic (febrile) malaria, malaria-associated symptoms, and anaemia were evaluated by uni- and multivariate analysis.

This work is part of their previous published work (REF 21). The authors should clearly explain in the objectives what is different in the present manuscript from REF 12 and 21, which justifies a separate publication of the same study, because there are several overlaps between their previous published papers and the present manuscript.

MAJOR COMMENTS:

Self-plagiarism, lines 79-95: This paragraph is almost identical to the Methods section (sub-section MTTT of the population”) of REF 21 (Ndong et al. 2019). Please re-word the entire paragraph.

Lines 78-95, Study design: The time schedule during the study period is not clearly described in this section. It’s only after reaching line 209 that I understood that (i) there was a two-week pre-MTTT period, (ii) of the four MTTT interventions, the first one was conducted in July 2017 and that this MTTT is referred to as the “baseline”; (iii) the months and year (2017 or 2018) in which two other MTTT interventions were implemented should be given; (iv) the fourth and last MTTT occurred in July 2018 and this is referred to as the “evaluation.” For clarity, please describe these steps in the study design.

I also found the implementation scheme of MTTT interventions in Ndong et al. 2019 (REF 21). As an alternative, the authors can state that the implementation scheme is detailed in their previous published work (REF 21) and cite this reference in the study design sub-section.

Lines 185-191 and Table 4: I find this table confusing, notably because there are two P-values for each characteristic. I did not follow what the text says (lines 185-191) with regards to the results in Table 4. Please clarify.

MINOR COMMENTS:

Lines 3-4, Article title, “…in a malaria endemic area in Ghana”: Are there non-malaria endemic areas in Ghana, or is the entire country endemic for malaria? If there is no malaria-free areas in Ghana, the title can be revised by deleting “…in a malaria endemic area.”

Line 32: …the most common presenting symptoms (or diagnosis)

Line 35, “new vaccines”: Please be more precise. Are the authors referring to malaria vaccines?

Line 36: …countries in sub-Saharan Africa

Line 44: …among children. (period, then a new sentence) However, non-malaria causes…

Line 56: …intermittent preventive treatment (IPT)… (the authors should also explain what IPTc stands for at the appropriate place in the main text)

Line 58: home-based

Line 60: Mass test, treat and track (MTTT) has also…

Line 60: I think that it is important to add the definition of MTTT for uninitiated readers.

Line 61: …[12]. (period)

Lines 64-65, “… such as chills and rigors, fever, cough, headache, vomiting and convulsion”: These are various symptoms seen in febrile illnesses. This part of the sentence can be deleted for clearer understanding.

Line 69: …a larger study the implementation of which involved delivery of MTTT…

Line 71: sub-district

Line 71: It would be helpful if the authors can add a figure (a map) showing the location of Pakro, Ghana.

Lines 71-72: Community entry activities…were conducted…

Lines 73, 214: What is “durbars”?

Line 79: positive malaria rapid diagnostic test (RDT)

Line 82, “Ndong et al., 2019”: Please use the correct reference number instead of “Ndong et al., 2019.” Is it REF 12 or 21?

Line 84: histidine-rich protein II (spelling of “histidine”; there is only one type of histidine-rich protein II – in singular form)

Line 85: P. falciparum (space between “P.” and “falciparum”)

Line 87: artemisinin-based combination therapies (ACTs) (I think that even if the abbreviation of “ACTs” was presented in the abstract, it should be re-introduced in the main text).

Line 90: “All children who tested positive for malaria parasitaemia were treated… (delete the commas)

Lines 91-92: Instead of “ACTs,” please specify which drug combination was used in this study. If both artesunate-amodiaquine and artemether-lumefantrine were used in the study, please explain how it was decided whether artesunate-amodiaquine or artemether-lumefantrine was given to an individual malaria-positive child. More details on treatment are necessary. Were all drug tablets given to the parents after the first dose? Who administered the drugs on days 1 and 2? The parents or community-based health volunteer (CBHV)? Was each drug intake supervised by the CBHV? If these details are explained in the authors’ previous works, the published work should be cited for further information.

Line 92: (artesunate-amodiaquine or artemether-lumefantrine).

Line 93: In children who vomited within five minutes…

Lines 96-99: What kind of specific information (age, sex, symptoms, etc.) was gathered through the questionnaire?

Line 99: use of insecticide

Line 102: were recruited, trained, and provided with

Line 103: There is something missing in this sentence “The CBHVs the, provide community-based timely testing of febrile cases.” Please correct it.

Line 107: to record all febrile cases two weeks before implementing MTTT

Lines 123-124, “Any child whose caretaker has immediate plan to relocate outside the study area has a life-threatening illness (excluding malaria)”: The meaning of this sentence is not clear. Please rewrite it.

Line 127, Ndong et al., 2019”: Please use the correct reference number instead of “Ndong et al., 2019.” Is it REF 12 or 21?

Line 137: haemoglobin (small letter “h”) (Hb)

Line 144: The chi-square test was used…

Lines 140-146: The authors used univariate and multivariate regression analysis, as stated in lines 166-167. This should be mentioned in the Methods section (under sub-section 2.9 data management and analysis).

Line 149: females

Line 152, “some of the children were not available”: Please give the exact number of children.

Lines 156-157, Table 1: The terms “baseline,” “at evaluation,” “baseline survey,” “and endline survey” are confusing. Please specify exactly when is the “baseline” and “evaluation” by either providing the month and year corresponding to each of these terms or specifying which of four MTTT the authors are referring to. This information can be added to the Table legend.

Line 157, “Table 2”: Should it be Table 1, instead of Table 2?

Line 159, “Table 2”: Should it be Table 1, instead of Table 2?

Line 160: a significant decrease…while the decrease in AP was not significant…

Line 162: …insecticide-treated net (ITN) decreased significantly…

Line 166: odds-ratio (OR)

Lines 167-168: Please provide the values of the mean body temperature in children with AP vs those without AP. These data are not in the tables.

Lines 174-176, “…reduced from 55.1% at baseline to 50.7% at evaluation, a reduction of 14.52%”): The difference between 55.1% and 50.7% is 4.4%. How did the authors obtain 14.52%? Please check.

Line 181: headache, 39.6% at baseline…

Table 3, “vomit”: vomiting

Line 185: Is it Table 4, instead of Table 5?; …at baseline (July 2017) [close parenthesis] and July 2018

Line 196, “1116”: Should be 116?

Line 198, “At evaluation, the proportion of severe anaemia dropped significantly from 7.0% at baseline to 4.4% at evaluation”: From what I see in the table, the proportion of severe anaemia in < 5 yrs decreased from 8.0% (not 7.0%) to 4.4%. Moreover, in the table the chi-square value given is 16.2 (not 15.8). Please check and specify that the authors are talking about the sub-group of children < 5 years old.

Line 199, “moderate anaemia dropped from 43.5% at baseline to 23.4% at evaluation (X2 = 13.7)”: Please specify that the authors are talking about children < 5 years old. Please also add the chi-square value to the table.

Line 201, 60,7% to 77,7%”: 60.7% to 77.7%

Lines 208-209: over time until it reached 77% at evaluation

Lines 211-212: a decrease in coverage

Line 213: …was very high among the caretakers in the present study (or in our study)

Line 219: …the effect of malaria MTTT complemented with…

Line 223: decreased by 33% at evaluation compared to the baseline (?)

Line 226: The World Health Organization (WHO) recommends…malaria diagnosis…

Line 231: there was an overall diminution…parasitaemia-positive cases

Line 232: “(from 230 (53. cases at baseline…)” Please correct it.

Line 236: …the finding is in contrast to earlier reports

Line 238: parasitaemia (spelling)

Line 244: the high prevalence… corroborates earlier findings…

Line 247: Prevalence of febrile illnesses and their impact on __?? Is there something missing after impact? Impact on what?

Line 255: [22,23,27-29]. (period) However, few studies…

Line 256: …the use of antimalarials protects…

Lines 263-264: ITN (instead of “insecticide treated nets”)

Line 280: …home-based

Line 290: true impact

Line 296: What is “HbM”?

Line 305: delete the comma after “that”

Lines 305-308: The authors recommend a shorter interval of MTTT, i.e. every two months, instead of every four months. Although this is feasible in a pilot study or clinical research funded to be conducted in a limited geographic area, this recommendation (MTTT every two months) is probably too costly for any government to finance at a national level. Any comments?

REFERENCES:

REF 3, 4, 7 and many other references: Please check the format for authors, especially for the last author of these cited articles (for example, REF 3 …de Mast, Q.).

Some journal names are written in their complete form, whereas others are in their abbreviated form.

REF 7: The journal name is missing.

REF 10: This reference is incomplete – journal name, volume, pages.

REF 10, 19, 26, 31: Article title – The first letters of each word are in capital letter.
